# Extract of *Aster koraiensis* Nakai Leaf Ameliorates Memory Dysfunction via Anti-inflammatory Action

**DOI:** 10.3390/ijms24065765

**Published:** 2023-03-17

**Authors:** Seung-Eun Lee, Saetbyeol Park, Gwi Yeong Jang, Jeonghoon Lee, Minho Moon, Yun-Jeong Ji, Ji Wook Jung, Yunkwon Nam, Soo Jung Shin, Yunji Lee, Jehun Choi, Dong Hwi Kim

**Affiliations:** 1Department of Herbal Crop Research, National Institute of Horticultural & Herbal Science (NIHHS), Eumseong 27709, Republic of Korea; parksb93@korea.kr (S.P.); janggy@rda.go.kr (G.Y.J.); artemisia@rda.go.kr (J.L.); jyj2842@korea.kr (Y.-J.J.); yoong0625@korea.kr (Y.L.); jehun@korea.kr (J.C.); kimodh@korea.kr (D.H.K.); 2Department of Biochemistry, College of Medicine, Konyang University, Gwanjeodong-ro 158, Soe-gu, Daejeon 35365, Republic of Korea; hominmoon@konyang.ac.kr (M.M.); yunkwonnam@gmail.com (Y.N.); tlstnzz@konyang.ac.kr (S.J.S.); 3Division of Biotechnology and Convergence, College of Cosmetics and Pharm, Daegu Haany University, Kyungsan 38610, Republic of Korea; jwjung@dhu.ac.kr

**Keywords:** *Aster koraiensis* Nakai, memory, cognition, B-cell lymphoma 2, choline acetyltransferase, amyloid, inflammation, Npy2r

## Abstract

*Aster koraiensis* Nakai (AK) leaf reportedly ameliorates health problems, such as diabetes. However, the effects of AK on cognitive dysfunction or memory impairment remain unclear. This study investigated whether AK leaf extract could attenuate cognitive impairment. We found that AK extract reduced the production of nitric oxide (NO), tumour necrosis factor (TNF)-α, phosphorylated-tau (p-tau), and the expression of inflammatory proteins in lipopolysaccharide- or amyloid-β-treated cells. AK extract exhibited inhibitory activity of control specific binding on N-methyl-D-aspartate (NMDA) receptors. Scopolamine-induced AD models were used chronically in rats and acutely in mice. Relative to negative controls (NC), hippocampal choline acetyltransferase (ChAT) and B-cell lymphoma 2 (Bcl2) activity was increased in rats chronically treated with scopolamine and fed an AK extract-containing diet. In the Y-maze test, spontaneous alterations were increased in the AK extract-fed groups compared to NC. Rats administered AK extract showed increased escape latency in the passive avoidance test. In the hippocampus of rats fed a high-AK extract diet (AKH), the expression of neuroactive ligand–receptor interaction-related genes, including Npy2r, Htr2c, and Rxfp1, was significantly altered. In the Morris water maze assay of mice acutely treated with scopolamine, the swimming times in the target quadrant of AK extract-treated groups increased significantly to the levels of the Donepezil and normal groups. We used Tg6799 Aβ-overexpressing 5XFAD transgenic mice to investigate Aβ accumulation in animals. In the AD model using 5XFAD, the administration of AK extract decreased amyloid-β (Aβ) accumulation and increased the number of NeuN antibody-reactive cells in the subiculum relative to the control group. In conclusion, AK extract ameliorated memory dysfunction by modulating ChAT activity and Bcl2-related anti-apoptotic pathways, affecting the expression of neuroactive ligand–receptor interaction-related genes and inhibiting Aβ accumulation. Therefore, AK extract could be a functional material improving cognition and memory.

## 1. Introduction

Alzheimer’s disease (AD) is a neurodegenerative disease that affects more than 50 million people worldwide [1]. AD, the most common disease among age-related progressive neurodegenerative disorders, is characterised by impairments of neuronal integrity and cognitive function [2,3,4]. AD is seemingly caused by several factors, including inflammatory cascades, reduced synthesis of the neurotransmitter acetylcholine (ACh), and amyloid-β (Aβ) deposition [5]. Because the brain has already suffered extensive damage including accumulation of amyloid plaques, neurofibrillary tangles, and neural loss at the diagnosis stage, early diagnosis is essential for identifying effective treatments to pause or slow the disease’s progression [6]. It is suggested that in traumatic brain injuries, the development of anti-oxidative and anti-inflammatory strategies for the management of damage could be a subject of research, while altered antioxidant systems and inflammation play critical role in the aetiology of neurodegenerative disorders [7,8].

Inflammation is related to various neurodegenerative diseases [9], and the notion that inflammation is related to activated microglia and astrocytes in AD pathogenesis has been supported by many studies [10,11]. An inflammatory response related to the accumulation of amyloid-β (Aβ) protein is a pathological hallmark of AD [12]. Nitric oxide (NO) is generated by the constitutive endothelial NO synthase (eNOS) and neuronal NO synthase (nNOS). Although at a physiological level, NO acts as an intercellular messenger molecule regulating vascular tone, platelet activation, and immune responses, thus serving as a neurotransmitter, a large amount of NO produced by inducible NOS (iNOS) can inhibit the activity of enzymes, resulting in potential cytotoxicity and pathogenic development [13,14]. Moreover, NO and reactive NO produced by microglia could act as neurotoxic agents in neurodegenerative diseases [15]. Therefore, certain anti-inflammatory therapies might be able to delay the progression of neurodegenerative diseases, such as AD.

Regarding memory and learning, it has been suggested that N-methyl-D-aspartate (NMDA) receptors, typically found in the cortical and hippocampal domain, are involved in the initiation steps. Furthermore, NMDA receptors are related to spatial and working memory tasks as well as long-term potentiation, which is a type of memory [16]. Regarding the cholinergic hypothesis in AD, the loss of cholinergic neurotransmitters, namely choline acetyltransferase (ChAT), is the most consistent alteration in the brain of patients with AD. In transgenic mice overexpressing AChE in the brain, progressive deterioration of cognition was evoked, supporting the role of ACh in memory [17,18]. Because ACh controls the neuronal response in the brain, this modulation plays a role in the essential mechanism underlying this complex behaviour [19]. 

An animal model of scopolamine-mediated cognition impairment mimics the activity blockade of the acetylcholine receptor, yielding the transient cognition loss seen in AD [20]. Moreover, 5XFAD transgenic mice are also valuable tools because the marked Aβ deposition observed in the subiculum and cerebral cortex are strongly related to AD with Aβ deposition in human brains and those of other animal models [21,22].

*Aster koraiensis* Nakai (AK), a perennial plant that belongs to the Asteraceae family, grows on the Korean peninsula, and AK leaves have also been approved by the Korea Ministry of Food and Drug Safety as an edible material. AK has been traditionally used as a folk medicine to cure strokes, venom poisoning, wounds, and sputum, according to the text of Donguibogam (described in the 17th century, Korea) [23]. The inhibitory activity of an Aster genus plant on COX-2 and iNOS was also reported [24]. Certain components of AK, such as eudesmane-type sesquiterpene glucosides, caffeoylquinic acids (CQAs), astersaponin Ⅰ, and polyacetylenes, reportedly have physiological activities, including neuroprotective effects on Parkinson’s disease [25,26,27,28,29]. Additionally, AK has been determined to play a variety of other roles, including preventing diabetic nephropathy, improving anti-nociceptive diabetic retinopathy, inhibiting retinal angiogenesis, healing wounds, lowering postprandial glucose, and inhibiting inflammation [30,31,32,33,34,35,36]. However, the effects of AK leaf on cognitive dysfunction remain unclear. Currently, AK is not used as a functional material to improve cognition or memory. 

This study investigated whether the extract of AK leaf (AK extract) displays anti-inflammatory activity at microglial and macrophage cells and binding capacity on NMDA receptors in an in vitro assay. Furthermore, we aimed to analyse the ability of AK extract to improve cognition or memory in two rodent models of scopolamine-induced cognitive dysfunction and measure its potential to inhibit Aβ accumulation in the brains of mice. 

## 2. Results 

### 2.1. AK Extract Had Anti-Inflammatory Activity In Vitro

The effects of the 70% ethanol extract of AK on NO production, cell proliferation, and TNF-α level were evaluated in LPS-treated BV2 and RAW264.7 cells, respectively. AK extract treatment (40–60 µg/mL, final concentration) significantly and dose-dependently inhibited NO production by 19.4–13.7 µM nitrite compared to the negative control (29.5 µM; Figure 1A, *p* < 0.05), showing cell proliferation of 104.9–112.4% compared to 92.5% in the negative control of LPS-treated BV2 cells (Figure 1B, *p* < 0.05). Furthermore, AK extract treatment (5–50 µg/mL) inhibited TNF-α production to 2574–3019 pg/mL compared to 3782 pg/mL in the negative control of LPS-treated RAW264.7 cells (Figure 1C, *p* < 0.05).

The effect of AK extract on inflammation-related protein expression was measured using a Western blot assay in LPS-treated RAW 267.4 cells. In the analysis, AK extract with a final concentration of 50–100 µg/mL was found to reduce cyclooxygenase (COX)-2 expression by 0.70–0.38 compared to 1.03 of the negative control (NC) group (Figure 2A, *p* < 0.05) and iNOS expression by 0.75–0.72 (Figure 2B, *p* < 0.05) compared to 1.12 observed in the NC group (0.0%). 

### 2.2. AK Extract Has Inhibitory Effects on the Expressions of Inflammatory Protein and the Production of P-Tau in Cells

The effects of AK extract on the expression of p-NF-κB and the production of p-tau in amyloid_1–42_ (Aβ) treated SH-SY5Y cells were evaluated. In the assay, AK extract at 1.25 µg/mL indicated the inhibitory effects on p-NF-κB expression (0.41) compared to NC (1.04) and on p-tau protein production (59.4 pg/mg protein) compared to NC (75.9 pg/mg protein), as shown in Figure 3A,B (*p* < 0.05). 

### 2.3. AK Extract Had In Vitro Binding Capacity on NMDA Receptor

The antagonistic effect of AK extract on NMDA receptors was also measured. In the experiment, 25–200 µg/mL of AK extract inhibited NMDA receptor binding by 15.9–44.5% in a dose-dependent manner (Figure 4A, *p* < 0.05). Furthermore, the reference compound CGS19755 at the concentration of 2.2–67 µg/mL showed an inhibitory effect on NMDA receptor binding by 4.1–56.3% (Figure 4B, *p* < 0.05).

### 2.4. AK Extract Improved Memory in Rats Chronically Treated with Scopolamine

The rats were administered a commercial diet of 0.14% AK extract (AKL group) or 0.42% (AKH group). To measure the effect of AK extract on the choline-related system, the hippocampal ChAT activity and ACh in all of the groups were analysed. The hippocampal ChAT level (0.8 ng/mL) in the negative control (NC) group decreased more than that in the normal group (1.7 ng/mL). However, all of the AK extract groups showed higher ChAT levels (2.4 ng/mL and 3.1 ng/mL) than that of the normal group. Moreover, the AKH group showed a higher value in hippocampal ChAT level than that of Donepezil (2.2 ng/mL) (Figure 5A, *p* < 0.05). The serum ACh values of the AKL and AKH groups (7.2 pg/mL and 7.6 pg/mL) which were lower values than that of the Donepezil group (9.1 pg/mL), showed increasing tendencies compared to that of the NC group (6.7 pg/mL) (Figure 5B, *p* < 0.05). These results suggest that AK extract plays a positive role in cholinergic signal transmission, possibly enhancing memory. 

The effects of AK extract on the expression of hippocampal Bcl2, BDNF, and ERK1/2 were analysed by using western blotting. Regarding other hippocampal proteins, NC rats showed lower Bcl2 expression tendency (0.83), a biomarker of apoptosis, than that of normal rats (1.01). However, Bcl2 expression in rats administered AK extract showed increasing tendency by 1.03 and 1.08. Additionally, the AKH group showed the same level of Bcl2 expression as that of the Donepezil group (1.22) (Figure 6A, *p* < 0.05). Regarding the relative expression of BDNF, the NC group, which was treated only with scopolamine, showed a lower level (0.74) than the normal group (1.21). In contrast, the BDNF levels of the AKL and AKH groups (0.91 and 0.90) which were lower than that of Donepezil group (1.16), showed increasing tendencies compared to that of the NC group (0.74) (Figure 6B, *p* < 0.05). Moreover, the hippocampal p-ERK/ERK ratio in the NC group (1.01)was higher than that in the normal group (0.42). AK reduced the p-ERK/ERK ratio by 0.62 and 0.66 compared to 1.01 in the NC and 0.87 in the Donepezil groups (Figure 6C, *p* < 0.05). Rats in the AKH group showed an increase in alterations of 28.6%, which was similar to that in the normal (29.9%) and Donepezil groups (29.1%) in the Y-maze test relative to the NC group (25.5%; Figure 7A, *p* < 0.05). In the passive avoidance test, the escape latency of NC rats decreased by 36.5 s, compared to those of the normal group (274.2 s) and Donepezil group (226.7 s), whereas the escape latency of the AKL and AKH rats showed increasing tendencies by 125.5 s–150.3 s, respectively (Figure 7B, *p* < 0.05).

In the microarray assay on the total hippocampal RNA of AKH rat vs. NC, 40 probes showed a fold change (FC) ≥ 1.5; the results are shown as a hierarchical clustering heatmap created using Z-scores of the normalised value of the KEGG pathway analysis (Figure 8A). Compared to NC, the genes expressed in the hippocampal tissue of AKH rats were significantly related to neuroactive ligand–receptor interaction (Figure 8D, Table 1 A, *p* = 0.00582613). The neuroactive ligand–receptor interaction-related genes expressed in the AKH group included Npy2r, Htr2c, and Rxfp1, as shown in Table 1A. Compared to NC rats, the fold changes in the hippocampal expression of Npy2r, Htr2c, and Rxfp1 in AKH rats were 2.57–2.59, 1.81, and −1.94, respectively. In addition, the microarray assay of the rat total hippocampal RNA of Donepezil (Do) vs. NC yielded 58 probes with a FC ≥ 1.5 (Figure 8B). Compared to NC, the genes expressed in the hippocampal tissue of the Donepezil rats were significantly related to neuroactive ligand-receptor interaction (*p* = 0.00308635), PI3K-Akt signalling pathway (*p* = 0.03017463), African trypanosomiasis (*p* = 0.03127858), and Malaria (*p* = 0.04276662), as shown in Figure 8D and Table 1B. The FC values of neuroactive ligand–receptor interaction-related genes in Donepezil rats, Npy2r, Chrm5, Trhr, and Rxfp1, were 2.20, 1.58, 1.60, and −2.12, respectively. In addition, genes related to the PI3K-Akt signalling pathway, Sgk1, Igf2, and Col1a2, were 1.65, −0.34, and −1.96, respectively, and only Sgk1 was upregulated. The African trypanosomiasis and malaria-related genes LOC689064 and Hbb-b1 were both downregulated (Table 1B). Furthermore, the microarray analysis of the total hippocampal RNA of normal (Nor) group vs. NC showed 228 probes with FC ≥ 1.5 (Figure 8C). A major gene expressed in the hippocampus of Nor group rats was related to metabolic pathway (Figure 8D).

### 2.5. AK Extract Improved Memory in Rats Acutely Treated with Scopolamine

In the learning test conducted on days 1 through 4 of the MWM test, the time to reach the escape platform by the rats in the NC group showed a minor decrease. Meanwhile, rats in the AKL, AKH, normal, and Donepezil groups showed a marked tendency to decrease. On the fourth day, the escape times of the normal, NC, AKL, AKH, and Donepezil groups were 29.5 ± 7.2 s, 42.1 ± 11.1 s, 27.7 ± 7.5 s, 28.8 ± 8.3 s, and 29.6 ± 8.9 s, respectively. Thus, the escape times of the groups treated with AK extract were reduced to nearly those of the normal and Donepezil groups (Figure 9A, *p* < 0.05). The result suggests that AK extract increases learning capacity in this acutely amnesia-induced rodent model.

For the memory potency test conducted on the fifth day of the MWM test, the platform was excluded, and the free swimming (FS) times of the animals were measured in the target quadrant for 60 s using EthoVision software. The FS time of the NC group (13.6 ± 2.1 s) was decreased compared to that of the normal group (19.1 ± 3.6 s), whereas the FS times of the AKL and AKH groups were increased to 18.4 ± 2.2 s and 18.2 ± 2.8 s, respectively. The results of the AKH group indicated the same values (19.1 ± 3.6 and 18.7 ± 3.5 s) as the normal and Donepezil groups (Figure 9B, *p* < 0.05). The result suggests that AK extract improves both long-term and spatial memory.

### 2.6. AK Extract Reduced Aβ Accumulation and Protected Neuronal Cells in the Brains of Transgenic Mice

The suppressive effects of AK extract on Aβ accumulation in the hippocampal subiculum and deep cortex in the AD mouse model were analysed using the TS-staining method and immunohistochemical assay using a 4G8 antibody. In TS analysis, the TS-reactive subiculum domain (%) of these two groups was 2.84 ± 0.73 (100% as the control group) and 2.28 ± 0.64 (80.4%), respectively. Meanwhile, the TS-reactive cortex domain values (%) of the control and AK extract groups were 0.60 ± 0.16 (100%) and 0.48 ± 0.16 (80.2%), respectively (Figure 10, *p* < 0.05). 

In an immunohistochemical assay conducted to visualise Aβ accumulation using the 4G8 antibody, the 4G8-reactive subiculum areas (%) of the control and AK groups were 20.2 ± 3.9 (100% as the control group) and 15.2 ± 4.2 (75.27%), respectively. Meanwhile, the 4G8-reactive cortical areas (%) of these two groups were 9.9 ± 2.9 (100%) and 7.4 ± 2.5 (75.29%), respectively (Figure 11, *p* < 0.05). The results of these two experiments confirmed that AK extract could inhibit the accumulation of Aβ in the hippocampal subiculum and the deep cortical layer. 

In the immunofluorescent staining method using NeuN antibodies on the subiculum of transgenic mice, the number of neuronal cells per mm^2^ in the wild type, control, and AK groups (5XFAD mice orally supplemented with AK extract 200 mg/kg) was 1392.0 ± 153.3 (2.49-fold of the control group), 558.1 ± 111.0 (1.00-fold), and 687.9 ± 143.1 (1.23-fold), respectively. Therefore, AK extract exhibited suppressive activity on the death of neuronal cells in the hippocampal subiculum (Figure 12, *p* < 0.05). These results suggest that AK extract reduces Aβ production and protects neuronal cells from Aβ toxicity in the brain of 5XFAD mice. 

### 2.7. Phenolic Components of AK Extract 

The phenolic composition of the AK extract is shown in Figure 13 and Table 2. The major phenolics were CQAs (isochlorogenic acid A and chlorogenic acid), which accounted for 65.983 and 54.539 mg/g extract, respectively. This result suggested that those major phenolics could affect AK extract’s in vitro and in vivo activities.

## 3. Discussion 

The present study verified that the AK extract reduces NO production in LPS-treated BV2 cells (microglial cells) and the expression of inflammatory proteins, such as COX-2 and iNOS, in LPS-treated RAW 264.7 cells in a dose-dependent manner. Microglia are involved in the inflammatory event in Alzheimer’s disease. In other words, microglial dysregulation and overactivation can evoke the inflammation-mediated neurotoxicity that characterises neurodegenerative diseases [9,12]. Elevated levels of microglial-derived cytokines and other mediators have been found in the AD brain [4,37,38]. Moreover, microglia-produced NO and reactive NO, which are mediated by cytotoxic cytokines, could act as neurotoxic agents in neurodegenerative diseases [15,39]. Activated microglia are also an important neurotoxic player in Aβ deposition, as they can release various inflammatory molecules, including cytokines, free radicals, and chemokines, which are related to Aβ production during the beginning and progression of AD [40]. Our results show that AK extract could inhibit the development of neurodegenerative states associated with microglial-induced inflammation of the brain. 

AK extract affected and inhibited the production of *p*-tau in amyloid_1–42_ (Aβ)-treated SH-SY5Y cells. Furthermore, we analysed whether AK extracts affected inflammation- and apoptosis-related proteins in amyloid_1–42_ (Aβ)-treated SH-SY5Y cells. The assay results showed that AK extract affected the expression of p-NF-κB. NF-κB is a transcription factor that plays crucial roles in inflammation and apoptosis; optimal induction of NF-κB target genes requires phosphorylation of NF-kB proteins [41]. Therefore, our results imply that AK extract could exhibit anti-inflammatory and anti-apoptotic actions by inhibiting protein expression.

Furthermore, AK extract showed suppressive activity in a dose-dependent manner on NMDA receptors. NMDA receptors, a subtype of glutamate receptors, are observed in the cortical and hippocampal domains and are critical to the early steps of memory and learning. These receptors are related to several memory tasks, such as spatial, working, and passive avoidance memory, as well as long-term potentiation [16]. NMDA receptors exert two opposing roles in the brain. They adjust vital steps to develop synaptic organisation and plasticity, while the overactivation of NMDA receptors can induce neuronal death in neuropathological states such as AD [42,43]. Complete blockage of NMDA receptors has been observed to hamper neuronal plasticity. Thus, both low and high activity of the glutamatergic system can yield malfunction [44]. According to the in vitro assay results of the suppressive activities of AK extract on NO release and the expression of inflammatory proteins (COX-2, iNOS, p-NF-κB), as well as its antagonistic effect on NMDA receptors, AK extract potentially improves cognition and memory in amnesia-induced animal models. 

In this study, the hippocampal ChAT activity of the AK extract-administered groups was significantly higher than that in all the other groups. Moreover, the ACh level in the serum of the AKH group was higher than that of the NC group. Inflammatory cascades, ACh and ChAT deficiency, extracellular deposition of Aβ, and abnormalities in tau protein are all considered hallmarks of AD [5,45]. A loss of cholinergic neurons is observed in brains affected by AD [46]. Coordination of ACh on the response of neuronal networks in the brain links cholinergic modulation to complex behaviour [19]. The loss of ChAT activity in cortical and hippocampal areas has been associated with the severity and duration of the disease [45,46,47]. The results of the present study suggest that AK may ameliorate the downregulation of the hippocampal ChAT and serum ACh levels, which is correlated with cognitive dysfunction in abnormal physiological states such as AD. 

In this study, the hippocampal expression of Bcl2, BDNF, and p-ERK1/2, factors related to memory or cognitive function, was analysed in rats chronically treated with scopolamine. The Bcl2 gene controls apoptosis and is expressed in the nervous system. Specifically, it represses the death of nerve cells and is markedly reduced during neuronal degeneration in AD [48]. In this study, Bcl2 expression in AK extract-administrated rats, especially the AKH group, increased compared to that of NC rats, where it was decreased by scopolamine treatment. BDNF in the cortex and hippocampus acts as a growth factor for the nervous system and is related to memory ability [49]. The BDNF level in the NC group was lower than those of the normal, AK extract, and Donepezil groups. Hock et al. reported decreased levels of BDNF in the hippocampus and partial cortex of AD subjects compared with the control group [47]. Furthermore, ERK1/2 plays a critical role in synaptic plasticity, learning, and memory in the hippocampus. Abnormal activation of ERK1/2 in the hippocampus may impair the function of the hippocampus and memory of patients with AD [50]. In this study, the hippocampal p-ERK/ERK ratio of the NC group increased more than that of the normal group. Moreover, the p-ERK/ERK ratio of the AK extract groups decreased compared to that of the NC group. Thus, AK extract administration ameliorated the increase in the p-ERK/ERK ratio observed in the NC group. The results of this study suggest that AK extract has a memory-improving effect in animal models. The mechanism of AK extract could include a cascade of these proteins related to the apoptosis pathway, memory, or learning in a memory-impaired condition. 

In addition, gene expression analysis using an RNA assay chip showed that genes expressed in the hippocampus of AKH-administered rats were significantly related to neuroactive ligand–receptor interaction, including Npy2r, Htr2c, and RXfp1. In the AKH rat hippocampus, gene expression of Npy2r (neuropeptide Y receptor Y2) was increased (2.57–2.59-FC) compared to that in NC rats. It is reported that Npy2r variation is involved in the modulation of iconic memory processes [51]. Expression of Htr2c (5-hydroxytryptamine receptor 2C, serotonin receptor 2C) increased to 1.81 FC. Htr2c is seemingly involved in depressive disorders [52], and its mRNA expression is abundant in the dorsal-ventral axis of the murine hippocampus [53]. However, the expression of RXfp1 (relaxin family peptide receptor 1) was reduced to −1.94 FC. RXfp1, one of the G-protein-coupled relaxin family receptors, is involved in stress responses, memory, and emotional processing. Immunoreactivity of RXfp1 was reduced in the parietal cortex of non-depressed AD patients, but RXfp3 level was upregulated in AD subjects with persistent depression [54]. In contrast, the genes expressed in the hippocampus of Donepezil-administered rats were related to neuroactive ligand–receptor interaction, including Npy2r, Chrm5, Trhr, and Rxfp1 with 2.20, 158, 1.60, and −2.12 FC, respectively. Chrm5 (M2 muscarinic acetylcholine receptor), which was elevated in the hippocampus of Donepezil-administered rats, is one of the receptors significantly reduced in mice, with deficiencies in activity-dependent BDNF compared to wild type mice. In addition, its expression is upregulated by antidepressant imipramine treatment [55]. Moreover, the PI3K-Akt signalling pathway, analysed from Donepezil-administrated rats, plays an important role in the progression or repression of AD [56]. We found that Sgk1 gene expression was upregulated in PI3K-Akt signalling. However, several genes, including the signalling molecule Igf2 and the extracellular matrix gene Col1a2, showed negative regulation in the PI3K-Akt signalling pathway [57]. In Donepezil-treated rat hippocampus, African trypanosomiasis, known as sleeping sickness, and malaria caused by *Plasmodium falciparum*, a pathogen, seem not to be directly related to improved AD [58,59].

Therefore, according to the results of gene expression in AKH-administered rats, it could be thought that AK extract exerts memory-enhancing effects via the action of genes such as Npy2r, Htr2c, and RXfp1, whereas Donepezil similarly acts on Npy2r, Chrm5, and genes related to the PI3K-Akt signalling pathway. 

Among the behaviour experiments of the present study, the spontaneous alterations percentage of the AKH-administrated group increased significantly compared to that of the NC group in scopolamine-treated rats, showing the same levels as in the Donepezil (positive control) and normal groups. The escape latency of the AKL- and AKH-administrated rats in the passive avoidance test increased 2.4- and 2.0-fold, whereas NC rats showed a lower tendency (1.0-fold) relative to normal and Donepezil rats (3.8-, 3.2-fold, respectively). Furthermore, in the MWM test with the scopolamine-treated mice model, the escape latencies (in sec) of the AK extract groups from the second to the fourth day were lower than those of the control (scopolamine) group. On the fifth day, the swimming time in the target area of AKH rats (19.0 s) was higher than that of NC animals (15.2 s). Because AD has distinctive features such as impaired neuronal integrity and cognitive function [2,3,4], the results of this behaviour test indicate that AK extract improved both spatial memory and learning ability. 

This work also examined whether AK extract affects Aβ aggregation and impairment of hippocampal neurogenesis using transgenic 5XFAD mice with five familial AD mutations. The TS method and immunohistochemistry analysis using the 4G8 antibody were used to analyse Aβ accumulation. Moreover, immunohistochemistry with NeuN antibody analysed the capacity of AK extract to protect neuronal cells in the subiculum domain of 5XFAD mice. The assays showed that the TS-reactive areas of the subiculum and cortex in the AK extract-administered transgenic mice showed lower values than those of saline-treated mice (control animals). The 4G8-reactive areas in the subiculum and cortex from the AK extract-fed rats were also reduced compared to those of the saline-fed control rats. However, the number of NeuN-antibody-reactive cells in the subiculum of the AK extract-administered mice was higher than that in the saline-treated control mice. The immunohistochemical assay on neurons demonstrated a reduction in cell death following treatment with AK extract in 5XFAD mice. These results imply that AK extract inhibited Aβ accumulation in a part (subiculum and cortex) of the transgenic mice brain but enhanced the viability of neuronal cells in the subiculum. Various inflammatory responses, including the extracellular accumulation of Aβ, are characteristic pathological features of AD. Because Aβ oligomer-induced neurotoxicity may be reversible, suppressing Aβ deposition could be an important strategy to cure AD [60]. In AD pathology, Aβ mediates glial cell activation, which in turn evokes neuronal loss via neuro-inflammatory responses [61]. Therefore, the results from the experiments in this animal model suggest that AK extract, which has anti-inflammatory activity and an antagonistic effect on NMDA receptors, could be a candidate material for protecting against memory dysfunction or AD via the inhibition of Aβ accumulation and protection of neuronal cells. 

Finally, analysis with high-performance liquid chromatography was conducted to estimate the components that contribute to the activity of AK extract. Caffeoylquinic acids (CQAs) were found to be the major phenolic substances in AK extract. The contents of isochlorogenic acid A (3,5-dicaffeoylquinic acid, 3,5-DCQA) and chlorogenic acid were 65.983 and 54.539 mg/g extract, respectively. It has been suggested that chlorogenic acid could improve cognitive functions to some degree, aiding in the performance of complex missions in human studies [62]. Previous studies on structure–activity relationships suggested that the caffeoyl group in the compound (CQA) is critical for suppressive activity. These CQAs were also reported to inhibit Aβ transformation into β-sheets [63]. CQA treatment ameliorated cognitive impairments in APP/PS1 mice by increasing the activation of certain signalling pathways [64]. Additionally, an intake of chlorogenic acids for six months may enhance the memory function of patients suffering from memory loss [65]. It is also reported that 3,5-DCQA could reduce PC12 cell toxicity induced by amyloid β sheets [66]. Therefore, it is considered that isochlorogenic acid A and chlorogenic acid, the major components of AK extract, could play important roles related to AK extract’s activity in terms of memory improvement.

## 4. Materials and Methods

### 4.1. Plant Resource and Extract Preparation

The *Aster koraiensis* Nakai (AK) leaves used in this study were collected and dried in May 2018 in Eumseong county, Chungcheongbuk province, South Korea. A specimen of the sample plant was deposited in the National Institute of Horticultural and Herbal Science (NIHHS), Eumseong, Republic of Korea (Voucher No. MPS003049). Powdered AK leaves (1000 g) were used for extraction with 70% ethanol at 85 °C in a refluxing apparatus and filtered. This procedure was conducted two times. Extraction solvents were eliminated through evaporation in a vacuum at 50 °C and lyophilisation. Finally, crude AK extract (228.24 g) was obtained and stored at −20 °C until used. 

### 4.2. Chemicals

Dulbecco’s Modified Eagle Medium (DMEM) and foetal bovine serum (FBS) were purchased from Thermo Fisher Scientific (Waltham, MA, USA). In addition, 3-(4,5-Dimethylthiazol-2-yl)-2,5-diphenyltetrazolium bromide (MTT), lipopolysaccharide (LPS), dimethyl sulfoxide (DMSO), penicillin & streptomycin (PEST), sulfanilamide, N-1-napthyl-ethylenediamine dihydrochloride (NED), L-glutamate, Trizma base, ethylenediaminetetraacetic acid (EDTA), all-trans retinoic acid (RA), Aβ protein fragment 1–42 (Aβ_1–42_), Donepezil, scopolamine, and thioflavin S (TS) solution were supplied by Sigma Aldrich (St. Louis, MO, USA). Anti-iNOS, anti-COX-2, anti-β-actin, p-NF-κB, extracellular signal-regulated kinase 1 and 2 (ERK1/2), p-ERK1/2, horse radish peroxidase (HRP)-conjugated anti-rabbit and anti-mouse IgG were purchased from Cell Signaling Technology (Denver, MA, USA). Moreover, antibodies against B-cell lymphoma 2 (Bcl2), as well as brain-derived neurotrophic factor (BDNF), were supplied by Abcam (Cambridge, UK). Anti-4G8 and anti-neuronal nuclei antibodies were obtained from BioLegend (San Diego, CA, USA) and Merck (Millipore Burlington, MA, USA), respectively. Western ECL substrate was supplied by Bio-Rad (Hercules, CA, USA). ELISA kits for analysing ChAT and acetyl choline were supplied by Elabscience (Houston, TX, USA) and BioVision Inc. (Milpitas, CA, USA), respectively. A rat phospho tau protein ELISA kit was obtained from MyBioSource (Houston, TX, USA). Phosphate-buffered saline (PBS) was supplied by WelGene (Gyeongsan, Korea). Furthermore, [^3^H]CGP39653 and scintillation cocktail were purchased from Perkin Elmer (Boston, MA, USA). Finally, CGS19755 reference and BDNF protein were provided by Tocris Bioscience (Bristol, UK) and Alomone Lab (Jerusalem, Israel). 

### 4.3. Cell Culture

To observe the effects of AK extract on cell viability and NO production, BV2 cells, microglial cells derived from a mouse brain, were proliferated in DMEM, including 5% FBS and 1% PEST at 37 °C and 5% CO_2_ atmospheric conditions. To measure the inflammatory protein expression, RAW 264.7 cells, a macrophage cell line originating from a mouse tumour, were incubated in DMEM, including 10% FBS and 1% PEST at the same atmospheric conditions. The SH-SY5Y cell line used to assess p-tau and the proteins in the inflammatory pathway was cultured in RPMI-1640 media containing 10% horse serum, 5% fetal bovine serum, and 1% penicillin and streptomycin.

### 4.4. Analysis of Inflammatory Activities in BV2 and RAW 264.7 Cells

AK extract dissolved in serum-free (SF) medium was administered to BV2 cells, placed in 2 × 10^5^ cells/well on a 48-well plate for 2 h, which was followed by treatment with LPS (0.5 µg/mL) for 24 h. Then, 50 µL of the supernatant from each well was reacted with 50 µL of 1% sulfanilamide and 50 µL of 0.1% NED. The nitrite levels of the sample reactant and standards for the calibration curve were measured at 520 nm. 

To observe the proliferation of BV2 cells treated with AK extract, the cells were reacted according to the protocol stated above. After the medium of the wells treated with AK extract was aspirated, the residual cells were treated with MTT reagent (0.6 mg/mL of the final concentration) for 1 h. The resulting formazan was dissolved and optically measured at 540 nm. 

To analyse the effect of AK extract on inflammatory protein expression, RAW 264.7 cells (2~3 × 10^5^ cells/mL) were bred in a CO_2_ incubator for 24 h and reacted with AK extract dissolved in an SF medium for 2 h. The plate was treated with LPS (0.5 µg/mL) for 24 h. The supernatants from the wells were used for analysis of TNF-α level by ELISA kit according to the manufacturer’s instructions. The remaining RAW 264.7 cells in the lysis buffer were centrifuged at 12,000 rpm and 4 °C for 25 min. The supernatant was used for Western blot analysis using anti-iNOS, anti-COX-2, and anti-β-actin (1:1000 dilution) as primary antibodies, and HRP-conjugated anti-rabbit, anti-mouse IgG (1:2000 dilution) as secondary antibodies. The expression of the proteins treated with Western ECL substrates was identified using a ChemiDoc™ Imaging System (Bio-Rad, Hercules, CA, USA). 

### 4.5. Analysis of Protein Levels Related to Inflammatory Pathway and p-Tau Inhibition Activity in SH-SY5Y Cells

To assess the effects of AK extract on p-Tau production and the levels of p-NF-κB, SH-SY5Y cells (1 × 10^5^ cells/well) were seeded into 6-well plates and incubated for 24 h. The cells were treated with 10 μM RA and 2 and 5 nM BDNF for 1 day. The AK extract was subsequently used to treat the cells for 2 h, followed by the addition of 0.1 or 0.3 μM Aβ_1–42_ in all wells except for the control wells. After 21–48 h, the supernatant in each well was aspirated, and the remaining cells were washed with ice-cold phosphate-buffered saline and lysed. The lysis buffer-treated cells were then collected and centrifuged at 13,201× *g* and 4 °C for 20 min. The levels of phosphorylated-nuclear factor-kappaB (p-NF-κB) and p-Tau in the supernatant were evaluated using western blot analysis and an ELISA kit, according to the manufacturer’s instructions, respectively. For western blot analysis, primary antibody (p-NF-κB) was diluted 1:1000, and HRP-conjugated anti-rabbit secondary antibody was diluted 1:2000. The expression of the proteins treated with Western ECL substrates was identified using a ChemiDoc™ Imaging System (Bio-Rad, Hercules, CA, USA).

### 4.6. Analysis of Binding Capacity on NMDA Receptor

To analyse the antagonistic effect of AK extract on NMDA receptors, rat cerebral cortex membrane homogenate (1 mg protein) was incubated with 5 nM [^3^H] CGP39653 and AK extract dissolved in a buffer containing 5 mM Tris-HCl (pH 7.7) and 10 mM EDTA-Tris at 4 °C for 60 min according to the method reported by Sills et al. [67]. Nonspecific binding was determined in the presence of 100 µM L-glutamate. The reactant was filtered through glass fibre filters (GF/B, Brandel Inc., Gaithersburg, MD, USA) under vacuum conditions and rinsed with an ice-cold buffer. The dried filters were counted for radioactivity using a scintillation counter (LS series, Beckman Instruments, Fullerton, CA, USA). CGS19755 reference was used for comparing the effect of AK extract. 

### 4.7. Analysis of the Effects on Scopolamine-Treated Animals

An experiment was conducted to ascertain whether AK extract has the potential to ameliorate cognitive dysfunction or memory impairment in a scopolamine-induced animal model. To set the memory impairment rodent model, scopolamine was used as the amnesia inducer and Donepezil as the positive control drug. Scopolamine is a cholinergic receptor antagonist used to induce abnormal cognitive function in experimental animals.

An intraperitoneal injection of scopolamine was administered to cause cholinergic malfunction and impair cognition in rats. Scopolamine has been shown to decrease ChAT activity, which plays an important role in Ach synthesis in the cortex of patients with AD [68]. Amnesia induced by scopolamine has been suggested as a model for studying dementia in cases where the cholinergic function is a doubted probable cause [69]; scopolamine (1 mg/kg, IP) was found to decrease the behaviour of animals in the Y-maze test [70]. Meanwhile, Donepezil has been reported to directly inhibit microglial activation induced by Aβ and partially ameliorate neurodegeneration and memory impairment [71].

To measure the effects of AK extract on animal behaviour, three tests were used: the Y-maze test, the passive avoidance test, and the Morris water maze (MWM) test. The Y-maze test is used as a method of observing immediate, short-term, spatial memory and alteration tasks, while the MWM test is used to observe spatial memory [48,70]. Moreover, the MWM test is a reliable tool for measuring spatial learning in rodents and is useful for investigating the correlation between hippocampal synaptic plasticity and NMDA receptor function [72]. The beneficial effects of natural methods in treating cognitive dysfunction, such as AD, have been previously evaluated with behavioural tests, including the MWM test [73,74]. Finally, the passive avoidance test is a behaviour test widely used to measure learning and memory [75].

#### 4.7.1. Experiments in Rats Chronically Treated with Scopolamine 

Seven-week-old male Sprague Dawley (SD) rats (supplied by DBL Co., Eumseong, Korea) were divided into the following five groups: normal, negative control (NC), low-AK extract diet (AKL), high-AK extract diet (AKH), and Donepezil (positive control, PC) (*n* = 10). The diets of the AKL and AKH groups were prepared from a commercial diet (Ziegler Bros Inc., Gardners, PA, USA) with 0.14% and 0.42% AK extract, respectively. The doses and administration period of AK extract in the diets of rats and the numbers of animals were determined according to a preliminary experiment or a previously described method [76]. The final doses of AK extract in AKL and AKH were 103.23 mg/day and 300.51 mg/kg/day, respectively. The rats in the normal, NC, and PC groups were fed a commercial diet. All rats were provided with feed and water ad libitum for 21 days. The PC group was intraperitoneally administered 1 mg/kg of Donepezil once each day for 21 days. All groups, except for the normal group, were intraperitoneally injected with scopolamine (1 mg/kg) in 0.9% saline once each day, while the normal group was treated with 0.9% saline for 21 days. The protocol on animal care and use was reviewed and approved by the Institutional Animal Care and Use Committee of the National Institute of Horticultural and Herbal Science (Approval number: NIHHS-2021-001, date: 21 January 2021). 

#### 4.7.2. Analysis of Biomarkers in Animals Chronically Treated with Scopolamine

The passive avoidance test was conducted from the 18th to the 20th day, and the latency time for escape from a light room to a dark room was measured on the 20th day. The Y-maze test (2nd) was executed on the 21st day, and the result (alteration, %) was calculated by: (spontaneous alternation/total entry − 2) × 100. The rats were sacrificed under CO_2_ on the 22nd day. The collected brain tissue and serum were stored at –80 °C. The activity of the hippocampal ChAT and the content of serum ACh were analysed using an ELISA kit. The expression of hippocampal Bcl2, BDNF, and p-ERK/ERK was analysed using Western blotting. Specifically, a Western blot assay was performed for the hippocampal proteins with anti-Bcl2, anti-ERK, anti-p-ERK1/2, anti-β-actin (1:1000 dilution), and anti-BDNF (1:500 dilution) as primary antibodies, and HRP-conjugated anti-rabbit and anti-mouse IgG (1:2000 dilution) as secondary antibodies. Expression of the proteins under Western ECL substrate was identified by ChemiDoc™ Imaging Systems (Bio-Rad, Hercules, CA, USA). 

#### 4.7.3. Gene Expression Assay in Animals Chronically Treated with Scopolamine

The gene expression in hippocampal tissues (*n* = 4) of Nor, NC, AKH, and Donepezil (Do) groups was analysed. Total RNAs isolated from the tissues were processed for cDNA synthesis, cRNA amplification, ss-cDNA synthesis, fragmentation & terminal labelling, and hybridisation prior to being used to analyse gene expressions using a microarray assay chip. The results of gene analysis are shown as fold change (FC) relative to the values of NC. The data were processed and normalised with a robust multi-average (RMA) method implemented in Affymetrix^®^ Power Tools (APT) using Affymetrix Rat Gene 2.0 ST Array (Affymetrix Inc., Santa Clara, CA, USA). The false discovery rate (FDR) was controlled by adjusting the *p*-value using the Benjamini–Hochberg test. Hierarchical cluster analysis was performed on a set of differentially expressed genes (DEGs) using complete linkage and Euclidean distance as a measure of similarity. Gene-Enrichment and Functional Annotation analysis for a significant list of probes was performed using KEGG (http://kegg.jp (accessed on 17 May 2022 & 10 March 2023)).

All data analyses and visualisation of differentially expressed genes were conducted using R 3.3.2 (www.r-project.org (accessed on 17 May 2022 & 10 March 2023)).

#### 4.7.4. Experiment in Mice Acutely Treated with Scopolamine 

The memory deficit model of mice acutely treated with scopolamine was also used for the MWM behaviour test. Male ICR mice (5 weeks old) were supplied by DBL Co. (Eumsung, Korea). The mice were divided into a normal (vehicle), negative control (scopolamine), AKL (112.7 mg/kg AK extract in saline + scopolamine), AKH (328.0 mg/kg AK extract in saline + scopolamine), and Donepezil (1 mg/kg Donepezil in saline + scopolamine; *n* = 10) groups, and the numbers of animals were determined according to a preliminary experiment. The doses and administration period of AK extract were determined according to the experiment with rats chronically administered scopolamine. The mice were administered AK extract orally and Donepezil intraperitoneally for 21 days. Additionally, from the 17th to the 21st day, mice were intraperitoneally administered scopolamine (1 mg/kg/day), and the MWM test was performed. On the 5th test day, a cognition test was conducted using EthoVision software (Noldus, Wageningen, The Netherlands) to measure swimming times in the target area. The protocol for animal experiments was approved by the Institutional Animal Care and Use Committee of Daegu Haany University (Approval No. DHU2021-075, date: 7 July 2021).

### 4.8. Analysis of the Effects of AK Extracts on Transgenic Mouse AD Model 

An experiment was conducted to ascertain whether AK extract has the capacity to prevent Aβ accumulation in transgenic AD model mice. We used Aβ-overexpressing 5XFAD mutagenic mice (Tg6799; Jackson Laboratory, Bar Harbor, ME, USA). The Tg6799 line, which expresses the highest levels of mutant APP, is the most studied of the three lines of 5XFAD. All mice were four months old. The tail DNA of young mice, born by mating a 5XFAD male mouse with a B6SJL/F1 female mouse, was genotyped and classified as Tg (5XFAD) and non-Tg mice (wild type). The groups consisted of wild type mice + saline, 5XFAD mice + saline, and 5XFAD mice + AK extract (*n* = 5). The AK extract group was orally administrated AK extract (200 mg/kg in saline) once each day for 21 days. Doses, frequency of administration, and period of AK extract for 5XFAD mice were determined according to a preliminary experiment. The 5XFAD (negative control) and wild type groups were administered only saline once a day for the same period. 

TS was used to analyse amyloid plaques with β-sheet formation [60]. Immunohistochemistry using the 4G8 and NeuN antibodies was performed in AK-treated 5XFAD mice. Seven sections (–2.8 mm region) from the bregma per mouse, allowing identification of the hippocampal subiculum and deep cortex layer, were collected at 120 μm intervals, analysed immunohistochemically with anti-4G8 antibody (1:2000), anti-NeuN antibody (1:100 dilution), and stained with 5 mg/mL TS solution. A fluorescent image of the stained area was obtained using a Carl Zeiss LSM 700 Meta Confocal Microscope (Carl Zeiss, Oberkochen, Germany), and Image J software (version 1.50i, National Institute of Health, Sacaton, AZ, USA) was used for image analysis. This animal study was approved by the Institutional Animal Care and Use Committee of Konyang University (Project identification code: P-20-30-A-01, date: 15 October 2020).

The time schedule of the three animal experiments is shown in Figure 14.

### 4.9. Analysis of the Phenolic Composition of AK Extract

A total of 10 mg of AK extract was dissolved in 5 mL of 70% ethanol and filtered through a membrane filter (0.45 μm, PTFE, Whatman Inc., Florham Park, NJ, USA). The phenolic composition of the extract was measured using high-performance liquid chromatography (HPLC: 1200 series, Agilent Technologies, Santa Clara, CA, USA) with Fusion hydro-RP column (250 × 4.6 mm, 5 μm, Phenomenex, Torrance, CA, USA) and UV-visible detector. The mobile phase was composed of 0.1% formic acid in acetonitrile (ACN) and 0.1% formic acid in water according to the following schedule: 0 min (2% ACN), 0–5 min (2–2% ACN), 5–12 min (2–5% ACN), 12–17 min (5–8% ACN), 17–65 min (8v30% ACN), 65–68 min (30–30% ACN), 68–78 min (30–50% ACN), 78–100 min (50–100% ACN), and 100–110 min (100–100% ACN). Eighteen phenolic standards (neochlorogenic acid, chlorogenic acid, 4-*O*-caffeoylquinic acid, 1,3-di-*O*-caffeoylquinic acid, vicenin-1, isochaftoside, luteolin 7-*O*-glucoside, isochlorogenic acid B, isochlorogenic acid A, apigenin 7-*O*-glucoside, isochlorogenic acid C, diosmetin 7-*O*-glucoside, linarin, luteolin, apigenin, diosmetin, eupatorin, and acacetin) were purchased from Cheungdu Biopurify Phytochemicals Ltd. (Chengdu, Sichuan, China).The injection volume of the sample, flow rate of the mobile phase, and detection wavelength of the detector were set to 10 μL, 1.0 mL/min, and 340 nm, respectively.

### 4.10. Statistical Analysis

Data from multiple experiments are represented as the mean ± standard deviation (SD). Statistical significance between tests or groups was analysed by ANOVA (one-way analysis of variance) and Duncan’s multiple range test at *p* < 0.05 using SAS software (version 9.4, SAS Institute Inc., Cary, NC, USA). The statistical significance of the expression data in the microarray assay was determined using an independent t-test and fold change; the null hypothesis was that no difference existed among the groups. 

## 5. Conclusions

The present study demonstrates that extract of *Aster koraiensis* Nakai leaf (AK), which has anti-inflammatory activity in cells and antagonistic effects on NMDA, can ameliorate memory deficits through the modulation of ChAT activity and anti-apoptotic activity related to Bcl2. Furthermore, the study showed that AK extract could affect gene expression related to neuroactive ligand–receptor interaction, such as Npy2r, Htr2c, and Rxfp1 in scopolamine-induced memory-impaired rats/mice as well as inhibition of Aβ accumulation and neuronal cell death in transgenic mice. The CQAs are most likely responsible for this function. In conclusion, AK leaf extract is a potential candidate to be used as a functional material for improving cognition or memory. 

## Figures and Tables

**Figure 1 ijms-24-05765-f001:**
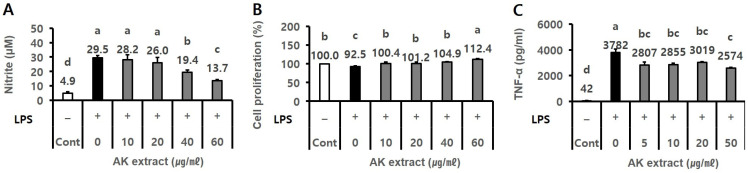
Effect of AK extract on (**A**) NO production, (**B**) cell proliferation in LPS-treated BV2 microglial cells, and (**C**) TNF-α level in LPS-treated RAW264.7 cells. Cells (2 × 10^5^ cells/mL for BV2, 3 × 10^5^ cells/mL for RAW264.7) seeded in a 48-well plate were treated with sample extracts for 2 h, and serum-free medium was used as the control. The wells were treated with 0.5 µg/mL LPS for 21–24 h. The optical density for the analysis of nitrite (*n* = 3), cell proliferation (*n* = 3), and TNF-α (*n* = 2) was measured at 520 nm, 540 nm, and 450 nm, respectively. (–), LPS-untreated experiment; (+), LPS-treated experiment; LPS, lipopolysaccharide; Cont, control; Statistical analysis was performed using SAS software. The values indicated with different letters were found to differ significantly at *p* < 0.05 by Duncan’s multiple range test.

**Figure 2 ijms-24-05765-f002:**
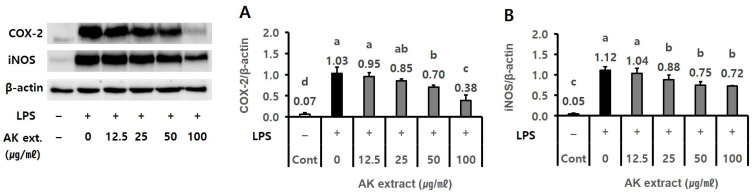
Effect of AK extract on the expression of (**A**) COX-2 and (**B**) iNOS in LPS-treated RAW 264.7 cells. RAW 264.7 cells (2 × 10^5^ cells/mL) were treated with sample extracts for 2 h, and serum-free medium was used as the control. The plate was treated with 0.5 µg/mL LPS for 24 h (*n* = 3 for Cox-2 and NOS). (–), LPS-untreated experiment; (+), LPS-treated experiment; LPS, lipopolysaccharide; Cont, control (*n* = 3). Statistical analysis was performed using SAS software. The values indicated with different letters were found to differ significantly at *p* < 0.05 by Duncan’s multiple range test.

**Figure 3 ijms-24-05765-f003:**
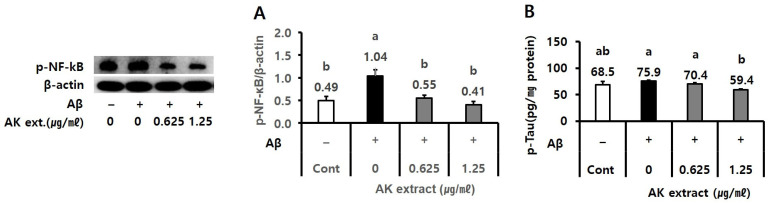
Effect of AK extract on the expression of (**A**) p-NF-κB, and (**B**) p-tau production in Aβ-treated SH-SY5Y cells. SH-SY5Y cells (1 × 10^5^ cells/mL) were treated with sample extracts for 2 h, and serum-free medium was used as the control. The plate was treated with 0.1 or 0.3 μM Aβ for 21–48 h. (–), Aβ-untreated experiment; (+), Aβ-treated experiment; Aβ, amyloid_1–42_; Cont, control (*n* = 2 for p-NF-κB and p-tau production). Statistical analysis was performed using SAS software. The values marked by different letters in A and B were found to differ significantly at *p* < 0.05 by Duncan’s multiple range test.

**Figure 4 ijms-24-05765-f004:**
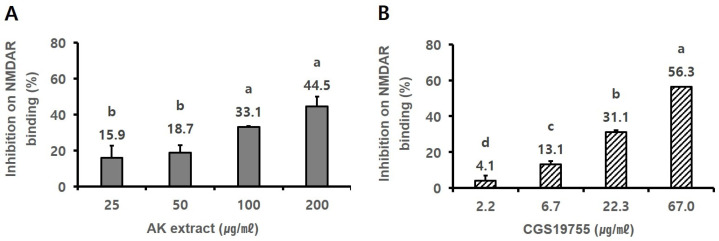
Inhibitory activities of control specific binding of (**A**) AK extract and (**B**) the reference compound (CGS19755) on NMDA receptors. The rat cerebral cortex membrane homogenate (1 mg protein) was incubated with 5 nM [3H] CGP39653 and AK extract in a buffer containing 5 mM Tris-HCl (pH 7.7) and 10 mM EDTA-Tris for 60 min at 4 °C. Nonspecific binding was determined in the presence of 100 µM L-glutamate. After incubation, the reactant was filtered through glass fibre filters in a vacuum and rinsed with an ice-cold buffer. The filters were dried and measured for radioactivity in a scintillation counter using a scintillation cocktail. Statistical analysis (*n* = 2) was performed using SAS software. The values indicated with different letters were found to differ significantly at *p* < 0.05 by Duncan’s multiple range test.

**Figure 5 ijms-24-05765-f005:**
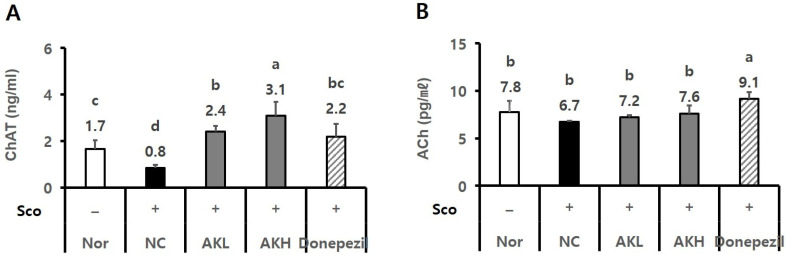
Effects of AK extract on the levels of (**A**) hippocampal ChAT activity and (**B**) serum ACh of rats chronically treated with scopolamine. The rats were randomly divided into five groups, as follows: Nor, normal, commercial diet + saline; NC, negative control, commercial diet + scopolamine in saline; AKL, a commercial diet with AK extract 0.14% + scopolamine in saline; AKH, a commercial diet with AK extract 0.42% + scopolamine in saline; Donepezil, commercial diet + Donepezil 1 mg/kg in saline + scopolamine in saline (*n* = 3 for ChAT assay and ACh assay). All rats, except those in the normal group, were intraperitoneally administered 1 mg/kg of scopolamine once daily for 21 days. Food and water were freely accessible. Statistical analysis was performed using SAS software. The values indicated with different letters were found to differ significantly at *p* < 0.05 by Duncan’s multiple range test.

**Figure 6 ijms-24-05765-f006:**
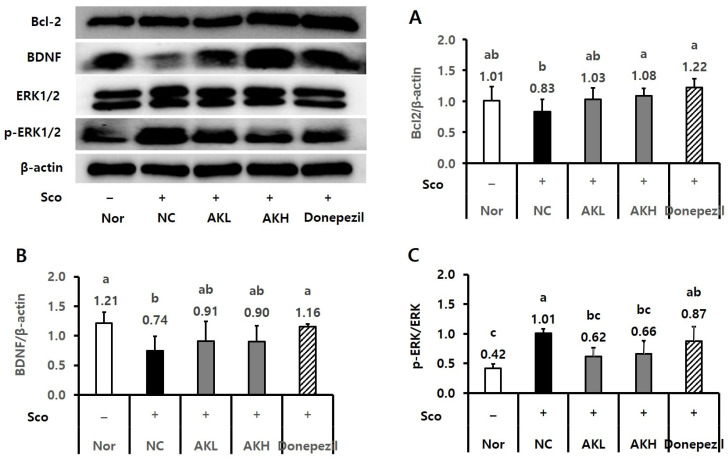
Effects of AK extract on the expression of hippocampal (**A**) Bcl2, (**B**) BDNF, and (**C**) p-ERK1/2/ERK in chronically scopolamine-treated rats. The rats were randomly divided into five groups: Nor: normal, commercial diet + saline; NC: negative control, commercial diet + scopolamine in saline; AKL: commercial diet of AK extract 0.14% + scopolamine in saline; AKH: commercial diet of AK extract 0.42% + scopolamine in saline; Do: commercial diet + Donepezil 1 mg/kg in saline + scopolamine in saline (*n* = 6 for Bcl2, *n* = 5 for BDNF and *n* = 4 for p-ERK/ERK). All rats, except those in the normal group, were intraperitoneally administered scopolamine (1 mg/kg/day) for 21 days. Food and water were freely accessible. Statistical analysis was performed using SAS software. The values indicated with different letters were observed to differ significantly at *p* < 0.05 by Duncan’s multiple range test.

**Figure 7 ijms-24-05765-f007:**
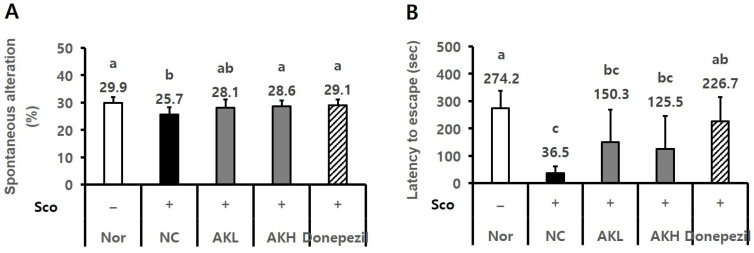
Learning and memory improvement effects of AK extract on chronically scopolamine-treated rats in (**A**) the Y-maze test and (**B**) the passive avoidance test. The rats were randomly divided into five groups: Nor: normal, commercial diet + saline; NC: negative control, commercial diet + scopolamine in saline; AKL: commercial diet of AK extract 0.14% + scopolamine in saline; AKH: commercial diet of AK extract 0.42% + scopolamine in saline; Donepezil: commercial diet + Donepezil 1 mg/kg in saline + scopolamine in saline (*n* = 8 for Y-maze and *n* = 6 for PAT). All rats, except those in the normal group, were intraperitoneally administered scopolamine (1 mg/kg/day) for 21 days. Food and water were freely accessible. Statistical analysis was performed using SAS software. The values indicated with different letters were observed to differ significantly at *p* < 0.05 by Duncan’s multiple range test.

**Figure 8 ijms-24-05765-f008:**
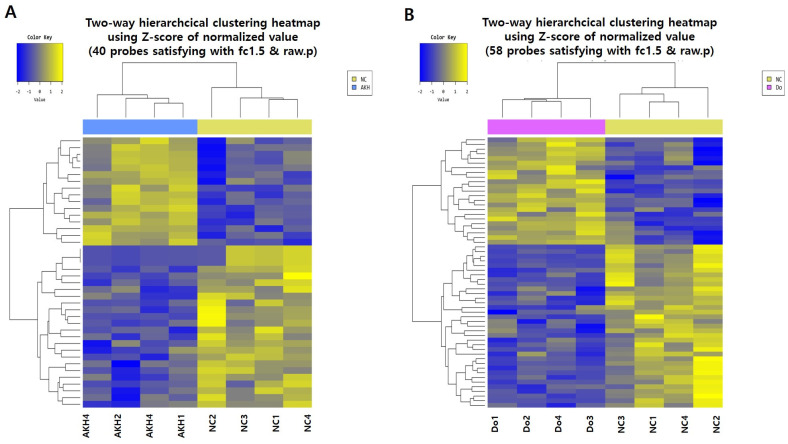
Effect of AK extracts on hippocampal gene expression in chronically scopolamine-treated rats using microarray assay. One-way hierarchical clustering heatmap of (**A**) AKH vs. NC, (**B**) Donepezil (Do) vs. NC, and (**C**) Nor vs. NC. (**D**) Dot plot image of differential pathways and data between AKH vs. NC, Donepezil vs. NC, and Nor vs. NC using KEGG pathway analysis. The rats were randomly divided into five groups: Nor: normal, commercial diet + saline; NC: negative control, commercial diet + scopolamine in saline; AKH: commercial diet of AK extract 0.42% + scopolamine in saline; Donepezil: commercial diet + Donepezil 1 mg/kg in saline + scopolamine in saline (*n* = 4). All rats, except those in the normal group, were intraperitoneally administered scopolamine (1 mg/kg/day) for 21 days. Food and water were freely accessible.

**Figure 9 ijms-24-05765-f009:**
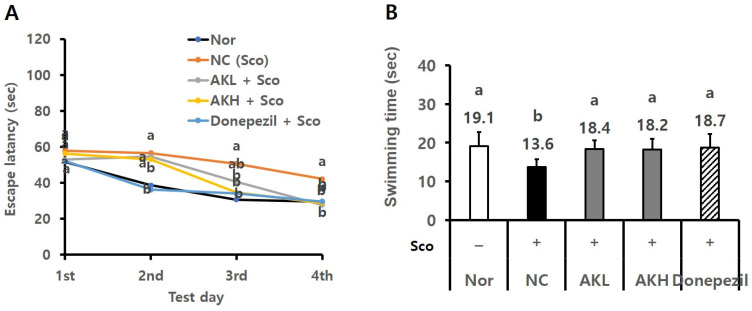
Effects of AK extract on behaviour in the Morris water maze test on acutely scopolamine-treated mice. (**A**) Escape latency in the acquisition phase, and (**B**) swimming time in the target quadrant in the probe trial. Mice were orally administered AK extract 112.7 mg/kg (low dose; AKL) or AK extract 328.0 mg/kg (high dose; AKH) for 21 days; after 30 min, they were intraperitoneally injected with scopolamine (1 mg/kg/day) for test days. Nor: normal, vehicle-treated; NC: negative control, scopolamine; Donepezil: Donepezil 1 mg/kg in saline + scopolamine (*n* = 10). Statistical analysis was performed using SAS software. The values indicated with different letters were observed to differ significantly at *p* < 0.05 by Duncan’s multiple range test.

**Figure 10 ijms-24-05765-f010:**
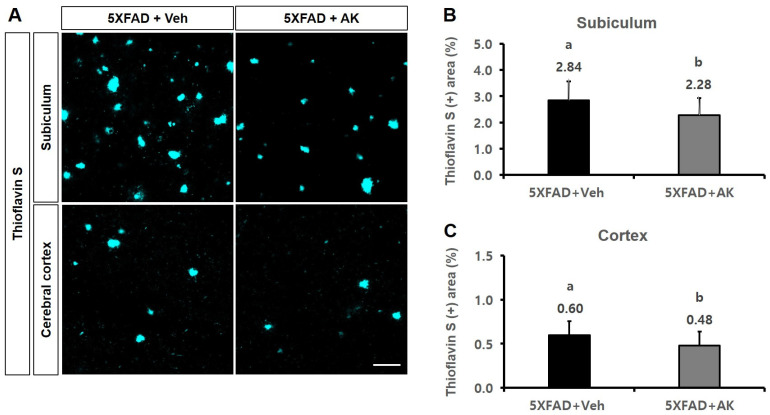
Inhibitory effects of AK extract on the β-amyloid accumulation in the subiculum and deep cortical layer in 5XFAD mice were analysed using the thioflavin S method. (**A**) Photographs of amyloid plaques in brain parts dyed with thioflavin S (TS); (**B**) intensity of the thioflavin S-positive area in the subiculum; (**C**) intensity of the thioflavin S-positive area in the cortex. Scale bar = 50 μm. 5XFAD + Veh: control group and 5XFAD + AK extract: AK extract group (*n* = 5). AK extract 200 mg/kg/day was orally administered for 21 days. Statistical analysis was performed using SAS software. The values indicated with different letters were observed to differ significantly at *p* < 0.05 by Duncan’s multiple range test.

**Figure 11 ijms-24-05765-f011:**
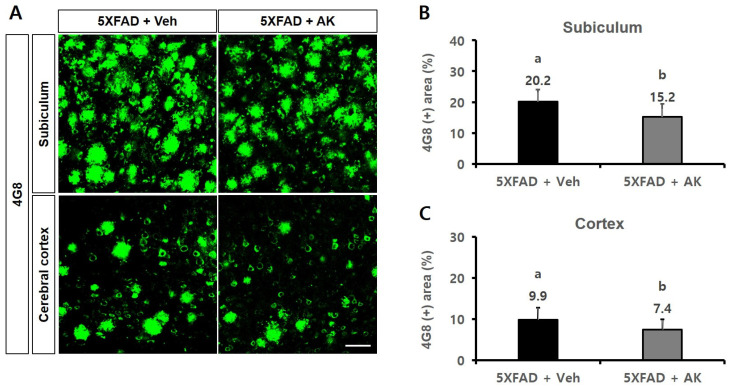
Inhibitory effects of AK extract on the β-amyloid accumulation in the subiculum and deep cortical layer in 5XFAD mice analysed by immunohistochemistry with 4G8 antibodies. (**A**) Photographs of amyloid plaques in brain parts dyed with 4G8 antibody; (**B**) Intensity of the 4G8 antibody-positive area in the subiculum; Scale bar = 50 μm. (**C**) Intensity of the 4G8 antibody-positive area in the cortex. 5XFAD +Veh: control group and 5XFAD + AK extract: AK extract group (*n* = 5). AK extract 200 mg/kg/day was orally administered for 21 days. Statistical analysis was performed using SAS software. The values marked by different letters were observed to differ significantly at *p* < 0.05 by Duncan’s multiple range test.

**Figure 12 ijms-24-05765-f012:**
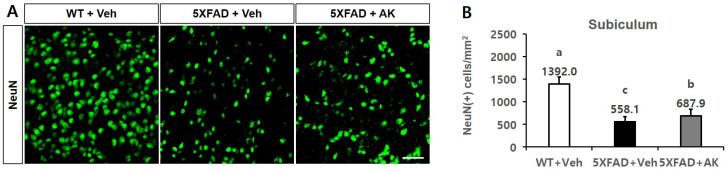
Inhibitory effects of AK extract on neuronal cell apoptosis of the subiculum observed by immunohistochemistry using NeuN antibodies. (**A**) Photographs of amyloid plaques in the subiculum dyed with NeuN antibodies. (**B**) Number of NeuN antibody-positive cells in the subiculum. Scale bar = 50 μm. WT + Veh: wild type normal group; 5XFAD + saline: control group; 5XFAD + AK extract: AK extract group (*n* = 5). AK extract 200 mg/kg/day was orally administered for 21 days. Statistical analysis was performed using SAS software. The values indicated with different letters were observed to differ significantly at *p* < 0.05 by Duncan’s multiple range test.

**Figure 13 ijms-24-05765-f013:**
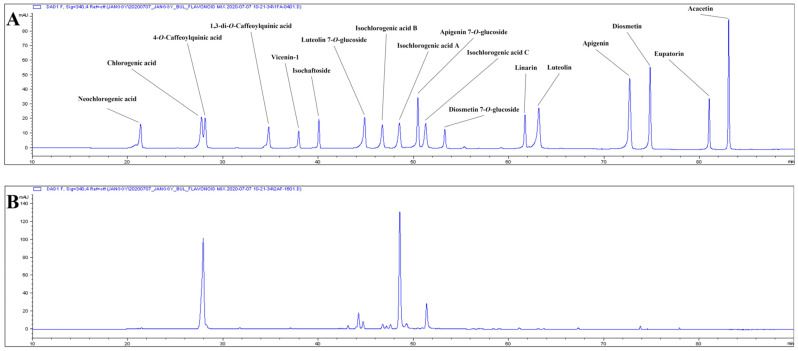
Typical chromatogram of AK extract. (**A**) Standard mixture, (**B**) AK extract. Standards: neochlorogenic acid, chlorogenic acid, 4-*O*-caffeoylquinic acid, 1,3-di-*O*-caffeoylquinic acid, vicenin-1, isochaftoside, luteolin 7-*O*-glucoside, isochlorogenic acid B, isochlorogenic acid A, apigenin 7-*O*-glucoside, isochlorogenic acid C, diosmetin 7-*O*-glucoside, linarin, luteolin, apigenin, diosmetin, eupatorin, and acacetin.

**Figure 14 ijms-24-05765-f014:**
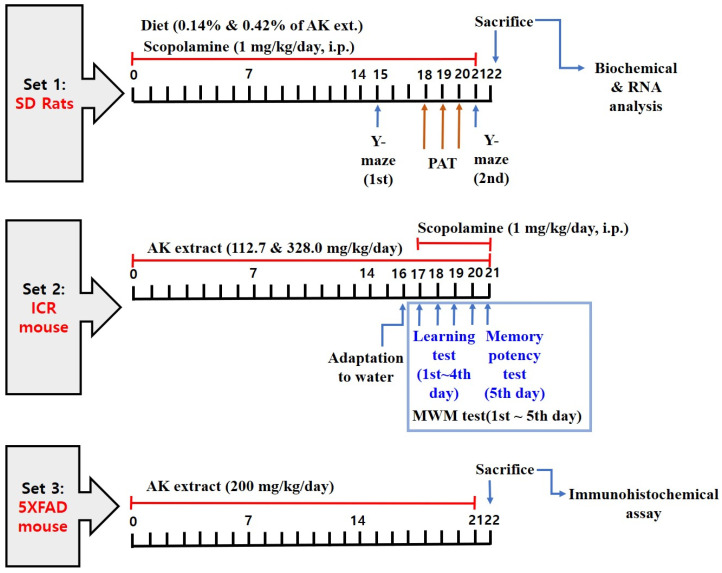
Schedule of animal experiments.

**Table 1 ijms-24-05765-t001:** Data of AKH vs. NC and Donepezil vs. NC from KEGG pathway analysis.

	MapName	*p* Value	Bonferroni	FDR	Gene Symbol	Fold Change	Raw *p* Value
(A) AKH vs. NC	Neuroactive ligand -receptor interaction	0.00582612	0.081565736	0.081565736	Npy2r	2.590162	0.023796771
		0.00582612	0.081565736	0.081565736	Npy2r	2.578688	0.036323573
		0.00582612	0.081565736	0.081565736	Htr2c	1.819431	0.048359603
		0.00582612	0.081565736	0.081565736	Rxfp1	−1.940098	0.048331939
(B) Donepezil vs. NC	Neuroactive ligand-receptor interaction	0.00308635	0.138885974	0.138885974	Npy2r	2.200639	0.04181401
		0.00308635	0.138885974	0.138885974	Chrm5	1.583830	0.0307853
		0.00308635	0.138885974	0.138885974	Trhr	1.605675	0.04482668
		0.00308635	0.138885974	0.138885974	Rxfp1	−2.124701	0.03873565
	PI3K-Akt signalling pathway	0.03017463	1	0.469178641	Sgk1	1.650502	0.01282808
		0.03017463	1	0.469178641	Igf2	−3.045132	0.01642387
		0.03017463	1	0.469178641	Col1a2	−1.967161	0.03576157
	African trypanosomiasis	0.03127858	1	0.469178641	LOC689064	−2.205598	0.01463965
		0.03127858	1	0.469178641	Hbb-b1	−1.572961	0.04611988
	Malaria	0.04276662	1	0.481124465	LOC689064	−2.205598	0.01463965
		0.04276662	1	0.481124465	Hbb-b1	−1.572961	0.04611988

**Table 2 ijms-24-05765-t002:** Phenolic composition of AK extract.

RT (min)	Phenolics	Content (mg/g extract, d.b.)
21.40	Neochlorogenic acid	0.544 ± 0.009
27.88	Chlorogenic acid	54.539 ± 0.938
28.25	4-*O*-caffeoylquinic acid	1.793 ± 0.042
40.11	Isochaftoside	0.112 ± 0.003
44.92	Luteolin 7-*O*-glucoside	2.895 ± 0.206
46.78	Isochlorogenic acid B	0.299 ± 0.010
48.58	Isochlorogenic acid A	65.983 ± 0.891
50.46	Apigenin 7-*O*-glucoside	0.377 ± 0.006
51.38	Isochlorogenic acid C	15.47 ± 0.521
63.16	Luteolin	2.752 ± 0.196

## Data Availability

Not applicable.

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
