# Peer review of "Extract of Aster koraiensis Nakai Leaf Ameliorates Memory Dysfunction via Anti-inflammatory Action"

_ijms, 2023, doi:10.3390/ijms24065765_

Round 1

Reviewer 1 Report

The authors investigated about the effects of Aster koraiensis Nakai (AK) leaf on cognitive dysfunction. Overall the topic could be interesting but some details could be improved.

 I recommend that the paper be accepted with minor revision:

a)  The authors should mentioned in the abstract more details about experimental models.

b)   In the introduction section, little previous evidence is provided about the importance of AD and other neurodegenerative disease in daily life. Incorporating comparisons with other studies would increase the strength of the paper. Please refer to doi: 10.1186/s13195-018-0346-2; 10.3390/antiox10111664; 10.4103/1673-5374.239448; 10.3390/antiox10050818.

c)   The authors should better emphasize the conclusions

d)  The authors should add the number of animals used in their study and how they choose the number. 

 e)  There are some minor grammar issues that should be fixed in order to aid the accessibility of the results to the reader.

Author Response

Thank you for your valuable comments.

Reviewer 2 Report

Summary: Natural products have been used a treatments for several diseases and health for numerous years. A perennial plant, Aster koraiensis Nakia, is no different. The leaves of this plant have been used as an edible material in Korea for use with stroke, wounds and the prevention of diabetic associated symptoms. Furthermore, some studies have shown that certain components extracted from this plant have neuroprotective effects against Parkinson’s Disease. While previous studies have identified these benefits of AK leaves and extract, no studies have investigated the used of AK for improving cognition and memory. This very impressive study identified AK extract as a potential therapeutic for Alzheimer’s Disease in vivo and in vitro in mouse models with memory impairments. This study identified AK extract as a potential therapeutic that can exhibit anti-inflammatory and anti-apoptotic characteristics. Exposure to the extract found that certain concentrations decreased the expression levels of anti-inflammatory and apoptotic proteins such as NF-KB, and caspase-8. They found that varying concentrations of AK extract decreased Ab accumulation and improved memory impairments, identifying this as a potential candidate for use in cognition and memory.  

Major Critiques: 

There are several sets of numbers that are presented in figures, raw data and normalized data; both values are not needed. However, there needs to be clarification on which values the statistics were completed on. Likewise, there should be consistency in which values are used throughout the manuscript. What values are the stats based on? The raw numbers or the absolute percentages? This needs clarification and consistencyIf for some reason there used on different numbers for different figures, this should be stated in the figure legend that corresponds.

There are several conclusions that are misleading or not correct in response to what the figures are reporting. The interpretation of such figures does not correspond with the statistical significance that is shown. Is the text right, or is the figure? Were the wrong letters used for the graph? This needs to be address and fixed.

All figure legends should contain the number of animals or replicates used for each treatment or specified if the same numbers were used to each. This should be addressed and primarily in figures 7-13, but also checked for in others. 

It would also be helpful for the figure panels in Figs 1-4 to be larger. It is difficult to see the information in them (for example, axes titles too small and different fonts). 

Moreover, the information at the top of each section in Figure 9 (A, B, C) is not visible and the KEGG pathway information in Figures D-F is blurry.

Figure 14 was confusing because the methods do not seem to be described in the manuscript and the results for this are not well described in the manuscript either. The figure legend states that Figure 14A is a “standard mixture”…of what? . This section of the manuscript should be revised and made clear for readers.

Minor Revisions

·      Line 379-378: AKL group did not show statistically higher ChAT levels than Donepezil according to the figure. The text says otherwise. 

·      Line 381: NC group not shown in Fig 6B

·      Line 401-402: AKH showed same level of BCL2 expression as Donepezil. 

·      Line 404-405: wrong interpretation. AKL and AKH share a letter in the figure so they’re not significant 

Figures suggestions:

·      Figure 2B: graph and legend say cell proliferation, however the text says cell viability. Which one is it? These are two different measurements 

·      Figure 4A: 6.25 à 0.625

·      Figure 4: panels A and B are opposite than the text 

·      Figure 4C/D: significances and text do not match

·      Figure 5A: y-axis wrong MNDAR à NMDAR

·      Figure 5B: x-axis doesn’t match reference compound in text 

·      Figure 7B: AKL, AKH and NC all share a letter so not significant. This differs from what the text reads 

·      Figure 9: Panels C and G are not discussed anywhere within the text 

·      Figure 11: Labels don’t match up with picture. “5XFAD+Veh” and “5XFAD+saline” 

·      Figure 12: No scale bar 

·      Figure 12: Labels don’t match up with picture. “5XFAD+Veh” and “5XFAD+saline” 

·      Figure 13: No scale bar 

Table Suggestions:

·      Table 1 (A) Why are the p-values, corrections and FDRs the same for each gene within the category? 

·      Table 1 (B) Why are the p-values, corrections and FDRs the same for each gene within each category?

·      Table 1 (B) Last three pathways aren’t discussed in text: why?

Author Response

Thank you for your valuable comments.
